# Effect of SARS-CoV-2 Vaccination on Symptoms from Post-Acute Sequelae of COVID-19: Results from the Nationwide VAXILONG Study

**DOI:** 10.3390/vaccines10010046

**Published:** 2021-12-30

**Authors:** Marc Scherlinger, Luc Pijnenburg, Emmanuel Chatelus, Laurent Arnaud, Jacques-Eric Gottenberg, Jean Sibilia, Renaud Felten

**Affiliations:** 1Rheumatology Department, Centre Hospitalier Universitaire de Strasbourg, 1 Avenue Molière, 67098 Strasbourg, France; luc.pijnenburg@chru-strasbourg.fr (L.P.); emmanuel.chatelus@chru-strasbourg.fr (E.C.); laurent.arnaud@chru-strasbourg.fr (L.A.); Jacques-eric.gottenberg@chru-strasbourg.fr (J.-E.G.); jean.sibilia@chru-strasbourg.fr (J.S.); renaud.felten@chru-strasbourg.fr (R.F.); 2Centre National de Référence des Maladies Auto-Immunes et Systémiques Rares, Est/Sud-Ouest (RESO), 67000 Strasbourg, France; 3Division of Rheumatology and Clinical Immunology, Beth Israel Deaconess Medical Center, Harvard Medical School, Boston, MA 02115, USA; 4Laboratoire d’ImmunoRhumatologie Moléculaire, Institut National de la Santé et de la Recherche Médicale (INSERM) UMR_S 1109, Institut Thématique Interdisciplinaire (ITI) de Médecine de Précision de Strasbourg, Transplantex NG, Faculté de Médecine, Fédération Hospitalo-Universitaire OMICARE, Fédération de Médecine Translationnelle de Strasbourg (FMTS), Université de Strasbourg, 1 rue Eugène Boeckel, 67084 Strasbourg, France; 5IBMC, UPR3572, CNRS, 2 allée Konrad Roentgen, 67084 Strasbourg, France

**Keywords:** long-COVID, vaccine, post-acute sequelae of COVID-19, SARS-CoV-2

## Abstract

Introduction: Few data are available concerning the effect of SARS-CoV-2 vaccination on the persistent symptoms associated with COVID-19, also called long-COVID or post-acute sequelae of COVID-19 (PASC). Patients and methods: We conducted a nationwide online study among adult patients with PASC as defined by symptoms persisting over 4 weeks following a confirmed or probable COVID-19, without any identified alternative diagnosis. Information concerning PASC symptoms, vaccine type and scheme and its effect on PASC symptoms were studied. Results: 620 questionnaires were completed and 567 satisfied the inclusion criteria and were analyzed. The respondents’ median age was 44 (IQR 25–75: 37–50) and 83.4% were women. The initial infection was proven in 365 patients (64%) and 5.1% had been hospitalized to receive oxygen. A total of 396 patients had received at least one injection of SARS-CoV-2 vaccine at the time of the survey, after a median of 357 (198–431) days following the initially-reported SARS-CoV-2 infection. Among the 380 patients who reported persistent symptoms at the time of SARS-CoV-2 vaccination, 201 (52.8%) reported a global effect on symptoms following the injection, corresponding to an improvement in 21.8% and a worsening in 31%. There were no differences based on the type of vaccine used. After a complete vaccination scheme, 93.3% (28/30) of initially seronegative patients reported a positive anti-SARS-CoV-2 IgG. A total of 170 PASC patients had not been vaccinated. The most common reasons for postponing the SARS-CoV-2 vaccine were fear of worsening PASC symptoms (55.9%) and the belief that vaccination was contraindicated because of PASC (15.6%). Conclusion: Our study suggests that SARS-CoV-2 vaccination is well tolerated in the majority of PASC patients and has good immunogenicity. Disseminating these reassuring data might prove crucial to increasing vaccine coverage in patients with PASC.

## 1. Introduction

It is estimated that 10–50% of patients infected with SARS-CoV-2 will continue to experience debilitating symptoms 12 weeks after their initial infection [1,2], a condition called long-COVID or post-acute sequelae of COVID-19 (PASC). The etiopathogenesis of PASC is uncertain and probably multifactorial, including viral- or immune-mediated organ injury, neurological involvement, dysautonomia, physical deconditioning and psychological burden [3,4]. Uncertainties concerning its pathogenesis have caused PASC patients to fear adverse effects from vaccination, prompting some not to take part in the first vaccination campaign [5]. In the case where a dysregulated immune response is at play in PASC, SARS-CoV-2 vaccination could worsen the symptoms. On the other hand, viral persistence due to defective anti-viral immunity has been hypothesized to account for PASC [6], suggesting that SARS-CoV-2 vaccination could potentially help restore viral immunity, improve symptoms burden and the potential impact of vaccination on other pathogenic mechanisms such as organ sequelae of previous infection or dysautonomia. These hypotheses support the importance of studying the impact of vaccination on PASC symptoms. Indeed, preliminary data suggest that SARS-CoV-2 vaccination could improve PASC symptoms [7]. The aim of this study was to evaluate the impact of SARS-CoV-2 vaccination on PASC burden.

## 2. Patients and Methods

We conducted an online survey (Google Form^®^, Santa Clara, CA, USA) among French-speaking adults recruited through social media platforms (i.e., Twitter^®^, San Francisco, CA, USA; Facebook^®^, Menlo Park, CA, USA) and patient associations (i.e., Après-J20).

The survey was anonymous, approved by an independent ethics committee (CE-2021-106) and all patients provided informed consent. Inclusion criteria were the definition of PASC by the French Haute Autorité de Santé [8]: a reported viral illness with a probable or confirmed COVID-19 diagnosis, persistent symptoms lasting >4 weeks and the lack of an alternative diagnosis to explain the presentation. The severity of a wide set of symptoms before and after vaccination was evaluated using a previously validated symptom set [9]. Information about the type of vaccine used or the reason for non-vaccination was evaluated. The questionnaire can be found in the Appendix A. At the time of the study, the vaccination scheme was considered complete if the patient reported 2 doses of vaccine or 1 dose of mRNA/ChAdOx1 vaccine with a prior biologically confirmed infection (either RT-PCR or serology). Quantitative data are reported as median with interquartile range (IQR 25–75) and qualitative results as a percentage. Quantitative data were compared using Student’s *t*-test, and qualitative data using the Chi2 test. Statistical analysis was conducted using JMP Software 14.0 (SAS Institute, Cary, CA, USA). A *p*-value < 0.05 was considered statistically significant.

## 3. Results

A total of 620 patients completed the survey between 3 and 17 August 2021, and 567 (91.5%) satisfied the inclusion criteria and were included in the analysis (Figure 1). The respondents’ median age was 44 (IQR 25–75: 37–50) and 83.4% were women. The initial infection was proven (with either RT-PCR, CT-scanner, serology or antigen test) in 365 patients (64%) and 5.1% had been hospitalized to receive oxygen. In total, 396 patients had received at least one injection of the SARS-CoV-2 vaccine at the time of the survey after a median of 357 (198–431) days following the initially-reported SARS-CoV-2 infection. Among them, 255 (64.4%) had a complete vaccination scheme, including 142 with two doses and 113 with one dose and prior positive RT-PCR or serology. Two patients received a combination of vaccines (ChAdOx1 followed by an mRNA vaccine) after the French health authorities recommended the use of ChAdOx1 only in patients above 55 years. Other patient characteristics are shown in Table 1.

Among the 380 patients who reported persistent symptoms at the time of SARS-CoV-2 vaccination, 201 (52.8%) reported an impact on symptoms following the injection. The impact of SARS-CoV-2 vaccination on PASC was not different depending on the vaccine used (*p* = 0.60). A global worsening of symptom severity was reported by 117 patients (31% of vaccinated PASC patients) and was mostly represented by fever/chills (74%), gastro-intestinal symptoms (70%), paresthesia (64%) and arthralgia (63%). Conversely, a global improvement was reported by 83 patients (21.8%) and was mainly driven by the improvement of anosmia (62%) and brain fog (51%). The vaccine impact on each symptom is shown in Figure 2. The vaccine impact on PASC symptoms lasted more than 2 weeks in 72.6% of patients reporting improvement and 63.7% of patients reporting worsening. The frequency of global improvement following SARS-CoV-2 vaccination was similar between virologically confirmed and non-virologically confirmed PASC patients (20.2% (48/238) vs. 24.3% (34/140), respectively, *p* = 0.35). However, non-virologically confirmed PASC patients were more likely to report symptom worsening following vaccination compared to the others (41.4% (58/140) vs. 24.8% (59/238) (*p* < 0.001)). After a complete vaccination scheme (see methods), a positive anti-SARS-CoV-2 IgG assay was reported by 93.3% (28/30) of initially seronegative individuals.

At the time of the study, 30% of PASC patients (170/567) remained unvaccinated. The characteristics of these patients were similar to vaccinated ones, except for a shorter delay between COVID-19 and survey completion (Table 1). The main reported reasons for postponing the SARS-CoV-2 vaccine were the fear of worsening PASC symptoms (55.9%) and the belief that vaccination was contraindicated because of PASC (15.6%).

## 4. Discussion

Our study suggests that more than two-thirds of patients with PASC may be vaccinated against SARS-CoV-2 without their symptoms worsening. Fever/chills and gastro-intestinal symptoms were the most frequently reported worsened symptoms, but are also commonly reported after SARS-CoV-2 vaccination in the general population [10]. Conversely, one out of five patients reported an improvement of their symptoms, mainly brain fog and anosmia which have been associated with disability in PASC. As previously shown [4], more than 90% of PASC patients report fluctuation of their symptoms which may account for some of the findings. Unexpectedly, PASC patients without confirmed initial COVID-19 were more likely to report symptom worsening following vaccination compared to confirmed ones (41.4% versus 24.8%, *p* < 0.001). This result could be explained by an increased nocebo effect in the population in which the belief of prior COVID-19 was present without biological proof [11]. Interestingly, the vast majority of patients (93.3%) who had a post-vaccinal serology reported having detectable anti-SARS-CoV-2 IgG, suggesting normal immunogenicity of the vaccine in this PASC population.

The characteristics of the vaccinated and non-vaccinated PASC populations were similar, except for the delay since the initial infection was shorter in the non-vaccinated population (Table 1). This was expected since French health authorities have recommended postponing SARS-CoV-2 vaccination at least 3 months after COVID-19.

The willingness to get vaccinated against SARS-CoV-2 in patients with PASC was mainly limited by the fear of PASC worsening, the belief that it is contraindicated in PASC and, as in the general population, the fear of adverse effects [12].

Our study has limitations. First, the recruitment was conducted using social media platforms that could select a younger population or one that is not accurately representative of the general PASC population. However, the age distribution of the PASC population was similar to that previously reported [4,13]. We believe that PASC patient recruitment using social media and belonging to a patient association (i.e., AprèsJ20) may also be a strength of the study. Indeed, patient association plays a major role in representing and defending patient interests and should be included in research studies. Interestingly, the involvement of patient involvement may help improve the dissemination and acceptability of its results [14]. In line with this, we studied patient-reported symptoms from a validated set of symptoms published to standardize the evaluation of PASC [9]. Second, the vast majority of patients received an mRNA-based vaccine, limiting the generalization of our findings to vector- or antigen-based vaccines. However, these data reflect the vaccines generally used in Europe or in the United States in this young population.

One difficulty in the study of PASC is its definition. In the present study conducted in mid-August 2021, we used the French health authority’s definition of PASC, while the WHO released a new definition of PASC on 6 October 2021. When comparing these definitions, they are almost identical: a probable or confirmed SARS-CoV-2 infection (both definitions), symptoms that last at least 2 months (WHO) or 4 weeks (French health authority’s definition), the absence of an alternate diagnosis (both definitions). In our study, the median duration of symptoms was 475 days (IQR:261–506) and no patients included in the analysis had symptoms <8 weeks. Finally, the aim of this study was descriptive and did not aim at comparing the safety of the SARS-CoV-2 vaccine between PASC and non-PASC individuals, explaining the lack of a control group to compare vaccination safety and reason for non-vaccination. Moreover, the limited number of included patients and the absence of information on comorbidities are also limiting factors. The conclusions based on this study are thus explorative and should be confirmed in wider studies taking these aspects into account.

## 5. Conclusions

Our study suggests that SARS-CoV-2 vaccination is well tolerated in the majority of PASC patients and has good immunogenicity. Disseminating these reassuring data might prove crucial to increasing vaccine coverage in patients with PASC.

## Figures and Tables

**Figure 1 vaccines-10-00046-f001:**
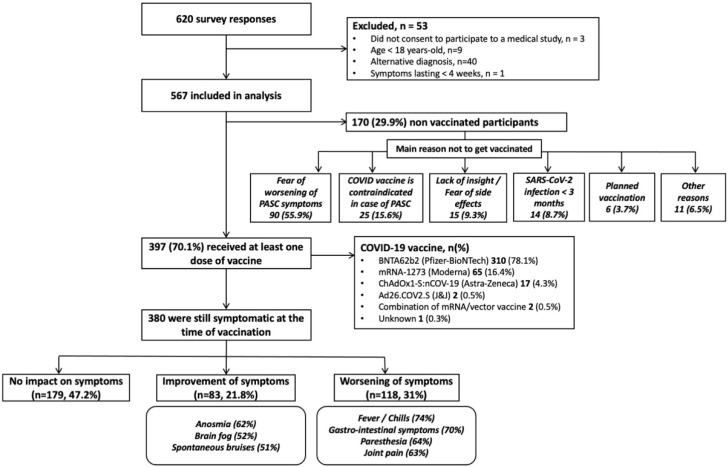
Study flow-chart and impact of SARS-CoV-2 vaccination on self-reported post-acute sequelae of COVID-19.

**Figure 2 vaccines-10-00046-f002:**
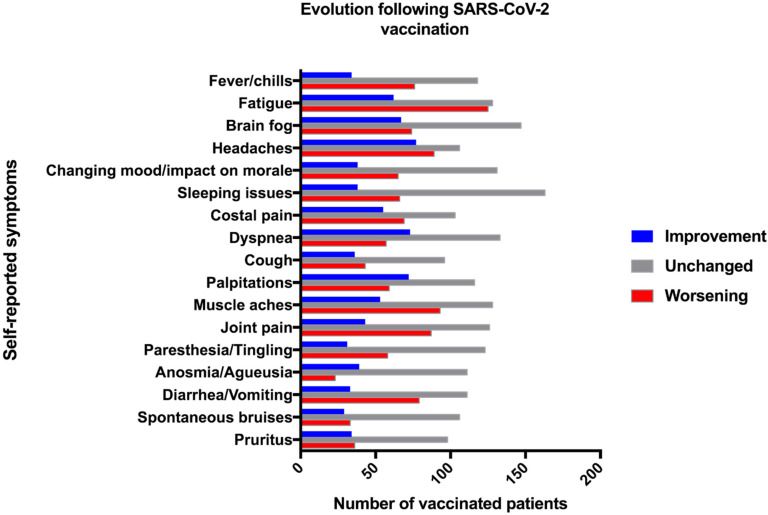
Evolution of self-reported symptoms following vaccination.

**Table 1 vaccines-10-00046-t001:** Characteristics from the included population. Lung CT, lung computed tomography; PASC, post-acute sequelae of COVID-19; RT-PCR, reverse transcriptase polymerase chain reaction.

	Total PASC Population (*n* = 567)	Vaccinated Population (*n* = 397)	Non-Vaccinated Population (*n* = 170)	*p*-Value Vaccinated vs. Non-Vaccinated
Female sex, % (*n*)	83.4% (473)	85.9% (327)	82.4% (146)	ns
Age, median (IQR)	44 (37–50)	44 (37–50)	42 (36–49)	ns
COVID-19 severity - Home-care - Hospitalized with oxygen therapy - Intensive care unit	94.9% (538)	95.3% (376)	94.7% (162)	ns
	5.1 (25)	4.7 (18)	5 3 (7)	
	0.7% (4)	0.8% (3)	0.6% (1)	
Confirmed COVID-19, % (*n*) - Positive RT-PCR - Positive lung CT - Positive serology - Positive antigen test	64.4% (365)	63% (250)	67.7% (115)	ns
	45% (255)	43.1% (171)	49.4% (84)	
	22.8% (129)	22.2% (88)	24.1% (41)	
	32.3% (183)	30.2% (120)	37.1% (63)	
	8.6% (49)	8.1% (32)	10% (17)	
Time since initial COVID-19, days, median (IQR)	475 (261–506)	483 (266–506)	325 (180–507)	**0.0066**
Number of persisting symptoms, median (IQR)	12 (9–15)	12 (9–15)	13 (10–15)	ns
Professional activity - Unchanged - Adapted to PASC - Interrupted	45% (255)	47.4% (188)	39.4% (67)	**0.016**
	18% (102)	19.4% (77)	14.7% (25)	
	37% (210)	33.3% (132)	45.9% (78)	
Vaccination - Time since initial infection - Number of Doses ○ One ○ Two	-	357 (198–431)	- -	-
		64.2% (255)		
		35.8% (142)		

## Data Availability

Data can be made available upon reasonable request to the corresponding author.

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
