# Peer review of "Effect of SARS-CoV-2 Vaccination on Symptoms from Post-Acute Sequelae of COVID-19: Results from the Nationwide VAXILONG Study"

_vaccines, 2021, doi:10.3390/vaccines10010046_

Round 1

Reviewer 1 Report

The brief report describes the results of a descriptive PASC survey in a post-vaccination patient population. The manuscript is well organized with statistical data and adequate references. In addition, the illustrations inform the analysis well.

Author Response

We thank the Reviewer for his positive evaluation of our work.

Reviewer 2 Report

The authors of “Effect of SARS-CoV-2 vaccination on symptoms from post acute sequelae of COVID-19: results from the nationwide VAXILONG study “ present a manuscript on self-reported symptoms coming under “long-COVID” in relation to vaccination. 

This brief report does not contain very much data, and the authors highlight the limitations of using social media to obtain responses from the general population.  For these reasons it is my opinion that as it is in its current state, this work falls short of the quality required for publication in Vaccines and would require considerably more data and work to improve. I have listed some comments for the authors to consider below, as well as line by line edits to improve the manuscript.

General comments:

The authors have not discussed the shortcomings of self-reported symptoms coming from social media sites. Social media sites are often the source of “fake news” and have contributed to the anti-vaccination movement which is a scourge to us all.  A major limitation is that users were recruited through social media rather than more respected organisations (hospitals, etc).

A major limitation is the lack of “controls”, i.e people without PASC who have completed the same survey and report any symptoms occurring.  This is important to put the PASC patients answers to the questionnaire in context of the rest of the population.  This may also explain why only 2/3 of patients do not report worsening of symptoms.  A 30% chance of “worsening symptoms” after vaccination is a very dangerous statistic to publish, and this manuscript doesn’t report enough data to make this statement.

There is no methods section, or any link to the questionnaire

Line 23, For an article on PASC, I would expect the authors to have proof read the manuscript before submission, a typo on the main subject of the paper on line 3 is not acceptable

Line 26, 620 with 567 completed, there is no reason to spell out 620

Line 27, “83.4% of respondents were women with a median age of 44”, this I think is a French to English syntax problem, which can sometimes alter the meaning of the sentence

Line 33, “a complete vaccination scheme”, are the authors suggesting two doses in this case? This may be confusing in the future when schemes will change (for example now with the booster doses)

Line 49, have the authors considered sampling patients without PASC to control for people who are against vaccination or finding excuses to avoid vaccination? As per general comment.

Line 53, the authors suggest that PASC patients need to “restore viral immunity”, however this is too strong an opinion to share given the lack of information and data on PASC.  The authors have suggested that viral persistence can account for PASC, but this is just one pathway and cannot support the opinion that viral immunity must be restored.

Line 77, are CT-lung scans used as a method of confirmation of COVID-19? This seems very non specific

Lines 74-83, copy paste from Abstract, this is not necessary, especially for a brief report

Line 85, figure 1 is coming before table 1, and table 1 is first cited in the manuscript

Line 86, table legends should be above the table

Figure 1, “comination” -> “combination”, please could the authors describe this combination in the text? I don’t believe combinations were authorized by regulatory authorities at the time this data would have been obtained?

Line 92-93, the authors should include data on the length that these symptoms occurred, as this is very relevant to “long COVID”. Is this what is meant by the sentence on line 97? 67.8% of patients reported these symptoms lasted over 2 weeks?

Line 104, how was this IgG assayed? There is no information on this in the MS

Author Response

Reviewer 2:

The authors of “Effect of SARS-CoV-2 vaccination on symptoms from post acute sequelae of COVID-19: results from the nationwide VAXILONG study “ present a manuscript on self-reported symptoms coming under “long-COVID” in relation to vaccination. 

This brief report does not contain very much data, and the authors highlight the limitations of using social media to obtain responses from the general population.  For these reasons it is my opinion that as it is in its current state, this work falls short of the quality required for publication in Vaccines and would require considerably more data and work to improve. I have listed some comments for the authors to consider below, as well as line by line edits to improve the manuscript.

We thank the Reviewer for his in depth review of our work. We agree with the Reviewer that this study has limitations. While we cannot change the initial design and methods of the study, we have implemented all the modifications suggested by the Reviewer and acknowledged the limitations accordingly in the revised manuscript.

General comments:

The authors have not discussed the shortcomings of self-reported symptoms coming from social media sites. Social media sites are often the source of “fake news” and have contributed to the anti-vaccination movement which is a scourge to us all.  A major limitation is that users were recruited through social media rather than more respected organisations (hospitals, etc).

We thank the Reviewer for raising this important point. We agree with the Reviewer concerning the difficulties inherent with the Social Media and its “fake news” which have hindered the voice of scientists and physician-scientist during the pandemic.

However, we believe that the fact that this study involves a population of PASC patients using social media may also be a strength of the study. Indeed, “anti-vaccination” individuals might be more prevalent in this included population, and the fact that we report reassuring data on vaccine safety in this same population might improve its acceptability and dissemination. The recruitment for this study was advertised by PASC patients association (AprèsJ20) and its results will also be advertised among patients, with the positive message that most patients have a neutral or positive effect of vaccination, and that the possibility of worsening of symptoms do exist but is likely to be linked by the vaccine effect (fever, pain) or by the natural history of PASC (which is characterized by fluctuating symptoms).

We have discussed the Reviewer point in the discussion section (page 6, first paragraph highlighted text): “First, the recruitment was conducted using social media platforms which could select a younger population or one that is not accurately representative of the general PASC population. However, the age distribution of the PASC population was similar to that previously reported [4,13]. Although the population may not e believe that PASC patient recruitment using social media and belonging to a patient association (ie. AprèsJ20) may also be a strength of the study. Indeed, patient association play a major role in representing and defending patient interests and should be included in research studies. Interestingly, the involvement of patient involvement may help improve the dissemination and acceptability of its results [14].“

A major limitation is the lack of “controls”, i.e people without PASC who have completed the same survey and report any symptoms occurring.  This is important to put the PASC patients answers to the questionnaire in context of the rest of the population.  This may also explain why only 2/3 of patients do not report worsening of symptoms.  A 30% chance of “worsening symptoms” after vaccination is a very dangerous statistic to publish, and this manuscript doesn’t report enough data to make this statement.

We thank the Reviewer for raising this point. The objective of this study was mainly descriptive and did not aim at comparing the safety of SARS-CoV-2 vaccine between PASC and non-PASC individuals. We agree with the Reviewer that the conclusions based on this study are explorative and should be confirmed in wider studies.

In accordance with the Reviewer’s point, we have discussed this point in the discussion section.

Discussion (page 6, last paragraph, highlighted text): “Finally, the aim of this study was descriptive and did not aim at comparing the safety of SARS-CoV-2 vaccine between PASC and non-PASC individuals, explaining the lack of control group. The conclusions based on this study are thus explorative and should be confirmed in wider studies.”

In the abstract, “Our study suggests that vaccination is well tolerated in the majority of PASC patients. “

There is no methods section, or any link to the questionnaire

We thank the Reviewer for raising this point. We tried to keep the method section short to keep the word count of a brief report. To address the Reviewer comment, we have added more details in the method section (page 2, highlighted text). We have also added a copy of the original questionnaire as supplementary material.

Line 23, For an article on PASC, I would expect the authors to have proof read the manuscript before submission, a typo on the main subject of the paper on line 3 is not acceptable

We thank the Reviewer for pointing out this typo that remained despite proofreading by the authors. We have corrected this typo accordingly and checked that there is no other PACS/PASC typo in the rest of the manuscript.

Line 26, 620 with 567 completed, there is no reason to spell out 620

We have modified the abstract and the main text accordingly.

Line 27, “83.4% of respondents were women with a median age of 44”, this I think is a French to English syntax problem, which can sometimes alter the meaning of the sentence

We thank the Reviewer for pointing out this syntax error. We have modified the sentence accordingly in the abstract and main text (highlighted in the revised manuscript).

Pages 1 and 2 (highlighted): “Respondents median age was 44 (IQR 25-75: 37-50) and 83.4% were women.”

Line 33, “a complete vaccination scheme”, are the authors suggesting two doses in this case? This may be confusing in the future when schemes will change (for example now with the booster doses).

We thank the Reviewer for raising this point. Complete vaccination scheme was defined as either two doses of mRNA or ChAdOx1-S:nCOV-19 (Astrazeneca) OR one dose of vaccine and a prior biologically proven infection (serology or RT-PCR). We have added this important point in the method section (page 2, highlighted text).

Added in the Methods section (page 2, highlighted text): “At the time of the study, the vaccination scheme was considered complete if the patient reported 2 doses of vaccine or 1 dose of mRNA/ChAdOx1 vaccine with a prior biologically confirmed infection (either RT-PCR or serology).“

Line 49, have the authors considered sampling patients without PASC to control for people who are against vaccination or finding excuses to avoid vaccination? As per general comment.

This in an interesting point. Unfortunately and as discussed earlier, our study was descriptive and did not include a control group to compare the reason for the absence of vaccination. Additionally, we could not find any published papers discussing this interesting aspect that we could have cited.

We have discussed the Reviewer point in the revised manuscript.

Discussion (page 6, last paragraph) : “Finally, the aim of this study was descriptive and did not aim at comparing the safety of SARS-CoV-2 vaccine between PASC and non-PASC individuals, explaining the lack of control group to compare vaccination safety and reason for non-vaccination.”

Line 53, the authors suggest that PASC patients need to “restore viral immunity”, however this is too strong an opinion to share given the lack of information and data on PASC.  The authors have suggested that viral persistence can account for PASC, but this is just one pathway and cannot support the opinion that viral immunity must be restored.

We completely agree with the reviewer that the pathogenesis of PASC is highly uncertain. On the one hand, immunopathologic mechanisms could be at play, which could be worsened by vaccination. On the other hand, viral persistence could be secondary to impaired viral clearance in the tissue, and vaccination could help improve the symptoms. Other pathogenic mechanisms could be involved (organ sequelae of initial infection, dysautonomia...), with unknown potential effect of the vaccination. These are only hypothesis to support the importance of studying the effect of vaccination in SARS-CoV-2. We do not favor any hypothesis, and our data do not allow us to state that any hypothesis is more likely than the other.

We have modified the introduction to explain all the hypothesis neutrally and explain the importance studying SARS-CoV-2 effect in PASC (page 2 highlighted text, and below).

“Uncertainties concerning its pathogenesis have caused PASC patients to fear adverse effects from vaccination, prompting some not to take part in the first vaccination campaign [5]. In case a dysregulated immune response was at play in PASC, SARS-CoV-2 vaccination could worsen the symptoms. On the other hand, viral persistence due to defective anti-viral immunity has been hypothesized to account for PASC [6], suggesting that SARS-CoV-2 vaccination could potentially help restore viral immunity and improve symptoms burden. Additionally, the potential impact of vaccination on other pathogenic mechanisms such as organ sequelae of previous infection or dysautonomia. These hypotheses support the importance to study the impact of vaccination on PASC symptoms.”

Line 77, are CT-lung scans used as a method of confirmation of COVID-19? This seems very non specific

The reviewer is correct. Many patients were initially diagnosed with COVID-19 during the early pandemic. At that time, testing capacities were scarce, and limited to severe cases. In some cases, lung CT-scan had been used as a surrogate to confirm the diagnosis. We agree with the Reviewer that this is less specific than the biological diagnosis tests. 20 patients (5.5% of confirmed COVID-19) had a COVID-19 diagnosis confirmed only by CT-scan. However, since the definition of PASC used in our study (as well as by the World Health Organization) does not necessitate a confirmed initial COVID-19, this approximation is unlikely to affect our results.

Lines 74-83, copy paste from Abstract, this is not necessary, especially for a brief report

We have modified the main text of the manuscript in order for it to be more informative and less redundant with the abstract. Please see highlighted text thorough the manuscript section.

Line 85, figure 1 is coming before table 1, and table 1 is first cited in the manuscript

This discrepancy has been corrected in the revised manuscript.

Line 86, table legends should be above the table

We have placed table legend above the table as suggested by the Reviewer.

Figure 1, “comination” -> “combination”, please could the authors describe this combination in the text? I don’t believe combinations were authorized by regulatory authorities at the time this data would have been obtained?

We thank the Reviewer for raising this point. We have corrected the figure accordingly. We have also added a sentence in the results section concerning this combination. Two patients received first the ChAdOx1 (Astrazeneca) vaccine followed by an mRNA vaccine (Pfizer, n = 1; Moderna, n = 1). As the Reviewer suggest, these combination were not recommended by the regulatory authorities at the time of the study. However, after several reports of vaccine-induced thrombotic thrombocytopenia (VITT) in young patients following the ChAdOx1 (Astrazeneca) vaccine, ChAdOx1 was subsequently reserved for patients older than 55 years old. Therefore, the patients who initially received one dose of ChAdOx1 had to receive a mRNA vaccine to complete their vaccination scheme. We believe this explanation to be true since the age of the two patients who received the vaccine combination were 49 and 53.

In accordance with the Reviewer comment, we have added information about this point in the main text.

Results (page 2 last paragraph, highlighted): “Two patients received a combination of vaccine (ChAdOx1 followed by an mRNA vaccine) after the French health authorities recommended the use of ChAdOx1 only in patients above 55 years. “

Line 92-93, the authors should include data on the length that these symptoms occurred, as this is very relevant to “long COVID”. Is this what is meant by the sentence on line 97? 67.8% of patients reported these symptoms lasted over 2 weeks?

Our questionnaire captured if the global impact of the vaccination on PASC symptoms (either improvement or worsening) lasted more or less than 2 weeks. To be more informative, we now show the duration of the improvement (72.7% lasted >2 weeks) and the worsening (63.7% lasted > 2 weeks) separately.

Concerning the duration of symptoms prior vaccination, we report the time since the initial COVID-19 (median 475 days, IQR [261-506]). We did not ask for the timing for each symptoms, as it would have made the questionnaire very difficult to complete.

Results section, page 4: “Vaccine impact on PASC symptoms lasted more than 2 weeks in 72.6% of patients reporting improvement, and 63.7% of patients reporting worsening.“

Line 104, how was this IgG assayed? There is no information on this in the MS

The patients were asked in the questionnaire if they had had a positive IgG assay for SARS-CoV-2 at least 2 weeks after their final dose of vaccine. This assay was conducted outside of the study and therefore we do not have direct information of the kit used for the assays. To take into account the Reviewer’s comment and be clearer for the reader, we modified the results and discussion section to make this point clearer.

Results section, page 4, last paragraph, highlighted: “After a complete vaccination scheme (see methods), a positive anti-SARS-CoV-2 IgG assay was reported by 93.3% (28/30) of initially seronegative individuals. “

Discussion section, page 5, first paragraph, highlighted: “Interestingly, the vast majority of patients (93.3%) who had a post-vaccinal serology reported having detectable anti SARS-CoV-2 IgG, suggesting normal immunogenicity of the vaccine in this PASC population. “

Reviewer 3 Report

The manuscript represents an analysis of a general questionnaire with no validation in which it is assumed that the conditions of the patients are represented by the answer. It can not be considered a proper study and the conditions related to vaccination may be partially appropriate. There are several other issues in the manuscript as the description of the symptoms does not take into account comorbidities, the use of medication, ie antihypertensive, insulin etc. The post covid or prorogued covid definition is inadequate see https://apps.who.int/iris/bitstream/handle/10665/345824/WHO-2019-nCoV-Post-COVID-19-condition-Clinical-case-definition-2021.1-eng.pdf

Finally, the discussion is poor and the references lacking. 

The manuscript could be suitable to another type of journal.

Author Response

Reviewer 3:

The manuscript represents an analysis of a general questionnaire with no validation in which it is assumed that the conditions of the patients are represented by the answer. It can not be considered a proper study and the conditions related to vaccination may be partially appropriate. There are several other issues in the manuscript as the description of the symptoms does not take into account comorbidities, the use of medication, ie antihypertensive, insulin etc. The post covid or prorogued covid definition is inadequate see https://apps.who.int/iris/bitstream/handle/10665/345824/WHO-2019-nCoV-Post-COVID-19-condition-Clinical-case-definition-2021.1-eng.pdf

Finally, the discussion is poor and the references lacking. 

The manuscript could be suitable to another type of journal.

We thank the Reviewer for the time he dedicated in evaluating our work.

The aim of this descriptive study was to evaluate the impact of SARS-CoV-2 vaccination on reported symptoms which are important for long-COVID patients such as fatigue, cephalalgia, brain fog... We agree with the Reviewer that this is neither a prospective cohort study nor a nationwide registry case-control study (ie, using Medicare registry). However, despite its limitation we believe that our study adds data to the literature, which remains very scarce on this subject.

Concerning the questionnaire, we used a validated set of symptoms for the evaluation of long-COVID criteria published by Tran et al. (reference 9). As we involved a patient association in the design and diffusion of our study, it seemed important to study patient-reported symptom burden.

Concerning the definition of post-acute sequelae of COVID-19 (PASC, or long-COVID), we conducted our study mid-August 2021, while the WHO definition of PASC was published October 6th 2021. When comparing the French and the WHO definitions, they are almost identical: a probable or confirmed SARS-CoV-2 infection (both definitions); symptoms that last at least 2 months (WHO) or 4 weeks (French health authorities definition); the absence of an alternate diagnosis (both definitions). In our study, the median duration of symptoms was 475 days (IQR:261-506), and no patients included in the analysis had symptoms < 8 weeks. We therefore conclude that we used a definition of PASC similar to the WHO definition. While we cannot change the methodology of our study after it has been conducted, we were able to discuss this point and compare the two definitions in the discussion of the revised manuscript.

To take into account the Reviewer’s comments, we have discussed these points in the discussion section, and further discussed our results in light of the current literature by adding appropriate references (11-12-13-14).

Added sentence in the discussion section (pages 5-6, highlighted in the manuscript):

Discussion section, page 5 first paragraph: “Unexpectedly, PASC patients without confirmed initial COVID-19 were more likely to report symptom worsening following vaccination compared to confirmed ones (41.4% versus 24.8%, p < 0.001). This results could be explained by an increased nocebo effect in the population in which the belief of prior COVID-19 was present without biological proof [11].”

Discussion section, page 6 second paragraph: “We believe that the fact that this study involved a population of PASC patients using social media and belonging to a patient association (ie. AprèsJ20) may also be a strength of the study. Indeed, patient association play a major role in representing and defending patient interests and should be included in research studies. Interestingly, the involvement of patient involvement may help improve the dissemination and acceptability of its results [14]. In line with this, we studied patient-reported symptoms from a validated set of symptoms published to standardize the evaluation of PASC [9]. “

Page 6, last paragraph: “One difficulty in the study of PASC is its definition. In the present study conducted mid-August 2021, we used the French health authorities definition of PASC, while the WHO released a new definition of PASC October 6th 2021. When comparing these definitions, they are almost identical: a probable or confirmed SARS-CoV-2 infection (both definitions); symptoms that last at least 2 months (WHO) or 4 weeks (French health authorities definition); the absence of an alternate diagnosis (both definitions). In our study, the median duration of symptoms was 475 days (IQR:261-506), and no patients included in the analysis had symptoms < 8 weeks.”

Page 6, last paragraph: “Finally, the aim of this study was descriptive and did not aim at comparing the safety of SARS-CoV-2 vaccine between PASC and non-PASC individuals, explaining the lack of control group. The conclusions based on this study are thus explorative and should be confirmed in wider studies. ”

Round 2

Reviewer 2 Report

The authors have made substantial edits to the manuscript, which is much improved

Author Response

We thank the Reviewer for his time and his positive assessment of our revised manuscript.

Merry and safe Christmas holidays.

Reviewer 3 Report

The manuscript was partially improved. In general, the comments and the amendments performed were appropriate and the inclusion of the questionnaire is important. However, there are issues concerning the interpretation of results based on the formula and the number of individuals which is a limiting factor. No inclusion of comorbidities in the questionnaire was also important. 

The authors should include the limitations of the study and future perspectives. 

Author Response

We thank the Reviewer for the time he dedicated in evaluating our revised manuscript as well as for his positive assessment of the amendments.

We agree with the Reviewers comments on limitations of the Study. In accordance with the Reviewer suggestion, we added a sentence in the last paragraph of the discussion to address these limitations and the need to address them in future studies.

Discussion, page 6, last paragraph: "Moreover, the limited number of included patients and the absence of information on comorbidities are also limiting factors. The conclusions based on this study are thus explorative and should be confirmed in wider studies taking into account these aspects."

We thank you again for your precious help in improving our work, and wish you safe and happy holiday season.

The coauthors